# Big Data to Knowledge Analytics Reveals the Zika Virus Epidemic as Only One of Multiple Factors Contributing to a Year-Over-Year 28-Fold Increase in Microcephaly Incidence

**DOI:** 10.3390/ijerph19159051

**Published:** 2022-07-25

**Authors:** Myriam Patricia Cifuentes, Clara Mercedes Suarez, Ricardo Cifuentes, Noel Malod-Dognin, Sam Windels, Jose Fernando Valderrama, Paul D. Juarez, R. Burciaga Valdez, Cynthia Colen, Charles Phillips, Aramandla Ramesh, Wansoo Im, Maureen Lichtveld, Charles Mouton, Nataša Pržulj, Darryl B. Hood

**Affiliations:** 1Department of Mathematics, College of Sciences, Antonio Nariño University, Bogotá 111321, Colombia; mpcifuentesg@unal.edu.co; 2Division of Environmental Health Sciences, College of Public Health, Ohio State University, Columbus, OH 43210, USA; 3Maestria en Salud Pública, Universidad Santo Tomás, Bogota 150001, Colombia; cmsuarez@ulibertadores.edu.co; 4School of Medicine and Health Sciences, Universidad Militar Nueva Granada, Bogotá 110111, Colombia; ricardo.cifuentesgarcia@gmail.com; 5Department of Computer Science, University College London, London WC1E 6BT, UK; n.malod-dognin@ucl.ac.uk (N.M.-D.); sam.windels@bsc.es (S.W.); natasa@cs.ucl.ac.uk (N.P.); 6Subdirectorate of Transmissible Diseases, Ministry of Health and Social Protection, Bogotá 110311, Colombia; direccion@colaboracionypragmatismo.org; 7Department of Family and Community Medicine, Meharry Medical College, Nashville, TN 37208, USA; pjuarez@mmc.edu (P.D.J.); wim@mmc.edu (W.I.); 8Department of Family & Community Medicine, School of Medicine, University of New Mexico, Albuquerque, NM 87106, USA; rovaldez@aol.com; 9Department of Sociology, College of Arts and Sciences, Ohio State University, Columbus, OH 43210, USA; colen.3@osu.edu; 10Electrical Engineering and Computer Science, University of Tennessee, Knoxville, TN 37996, USA; cphill25@vols.utk.edu; 11Department of Biochemistry, Cancer Biology, Neuroscience and Pharmacology, Meharry Medical College, Nashville, TN 37208, USA; aramesh@mmc.edu; 12Department of Environmental and Occupational Health, University of Pittsburgh, Pittsburgh, PA 15261, USA; mlichtve@pitt.edu; 13Department of Family Medicine, College of Medicine, University of Texas Medical Branch, Galveston, TX 77555, USA; cmouton@utmb.edu

**Keywords:** environmental public health, chemical and non-chemical stressors, Zika Virus (ZIKV), Big Data to Knowledge (BD2K), *public health exposome* (PHE), comparative network inferential analysis, microcephaly, agrochemicals, vulnerable populations

## Abstract

During the 2015–2016 Zika Virus (ZIKV) epidemic in Brazil, the geographical distributions of ZIKV infection and microcephaly outbreaks did not align. This raised doubts about the virus as the single cause of the microcephaly outbreak and led to research hypotheses of alternative explanatory factors, such as environmental variables and factors, agrochemical use, or immunizations. We investigated context and the intermediate and structural determinants of health inequalities, as well as social environment factors, to determine their interaction with ZIKV-positive- and ZIKV-negative-related microcephaly. The results revealed the identification of 382 associations among 382 nonredundant variables of Zika surveillance, including multiple determinants of environmental public health factors and variables obtained from 5565 municipalities in Brazil. This study compared those factors and variables directly associated with microcephaly incidence positive to ZIKV and those associated with microcephaly incidence negative to ZIKV, respectively, and mapped them in case and control subnetworks. The subnetworks of factors and variables associated with low birth weight and birthweight where birth incidence served as an additional control were also mapped. Non-significant differences in factors and variables were observed, as were weights of associations between microcephaly incidence, both positive and negative to ZIKV, which revealed diagnostic inaccuracies that translated to the underestimation of the scope of the ZIKV outbreak. A detailed analysis of the patterns of association does not support a finding that vaccinations contributed to microcephaly, but it does raise concerns about the use of agrochemicals as a potential factor in the observed neurotoxicity arising from the presence of heavy metals in the environment and microcephaly not associated with ZIKV. Summary: A comparative network inferential analysis of the patterns of variables and factors associated with Zika virus infections in Brazil during 2015–2016 coinciding with a microcephaly epidemic identified multiple contributing determinants. This study advances our understanding of the cumulative interactive effects of exposures to chemical and non-chemical stressors in the built, natural, physical, and social environments on adverse pregnancy and health outcomes in vulnerable populations.

## 1. Introduction

### Understanding the Complex Associations between a Zika Virus (ZIKV) Outbreak, Environmental Exposure to Chemical and Non-Chemical Stressors, and ZIKV-Related and Non-Zika-Related Microcephaly

On 1 February 2016, by stating that “Zika virus and its complications such as microcephaly and Guillain-Barré syndrome represent a new type of public health threat with long-term consequences for families, communities and countries”, the World Health Organization (WHO) declared a Zika epidemic in Latin America to be a Public Health Emergency of International Concern because it coincided with an unusual increase of clusters of microcephaly [1]. From fewer than 200 cases a year in Brazil, the incidence of microcephaly grew to 5734 cases in 2016 (Figure 1). Since 2013, when a large Zika outbreak began in French Polynesia, studies have suggested an association between Zika infection and congenital malformations and neurological and autoimmune complications [2,3] Ecological and individual studies completed during 2015–2016 were strongly suggestive of Zika’s causal relationships with these conditions [4,5,6,7]. However, as the 2016 spatial distributions of Zika infections and microcephaly among and within countries did not align, doubts emerged as to whether or not Zika was the single factor responsible for microcephaly. By 30 June 2016, only 25% of 61 countries and territories where WHO reported the ongoing transmission of Zika by epidemiological criteria and/or laboratory confirmation had associated cases of microcephaly and other neurological malformations [1,8]. Additionally, despite the Zika epidemic affecting neighboring countries, 95.3% of microcephaly cases were concentrated in Brazil. Inside the country, the burden of Zika-related microcephaly disproportionately affected the northeast states of Pernambuco and Bahia [9,10]. Therefore, some public health actions, including the use of larvicides [11,12,13,14] and vaccinations [15], as well as possible environmental, socioeconomic, or biological factors [16,17,18], emerged as likely alternative explanations for the uneven microcephaly distributions. By technical assessments, the WHO began to dispel some of these associations [19].

It is, however, well appreciated that Zika infection transmitted as a vector-borne disease can encompass multiple variables/factors in the complex interaction of people, vectors, and infectious agents within coexisting natural or anthropogenic, social, and individual ecosystems. Similarly, microcephaly itself comprises complex features linked to multiple causes. Before the Zika-related clusters, the rare cases of microcephaly in Brazil [20] were explained by complex genetic syndromes, environmental teratogens, low birth weight [21], and congenital toxoplasmosis and cytomegalovirus [22,23]. In Brazil’s Zika-related microcephaly clusters, multiple factors may have converged to shape the observed heterogeneity of the cases of mother-to-offspring transmission that arises from Zika’s mosquito inoculation and sexual transmission.

The aim of the present study was to understand how simultaneous and interdependent factors interacted to determine the incidence of Zika-related microcephaly. Our social ecological model for consideration of the involvement of social determinants of health (SDH) was informed by our public health exposome (PHE) [24,25,26] framework and tool chain that grounded the modeling of the complex Zika-related microcephaly clusters in Brazil. With regard to SDH, the WHO qualitatively classifies factors that determine health on three hierarchical scales of ‘socioeconomic and political context and social structure’, which include income, sex/gender, education, race/ethnicity, and ‘intermediary’ factors that directly condition health results, such as biological, psychosocial, and ecological/exposure determinants and the health care system [25]. The Ohio exposome study [27] and the present approach have guided the application of our Big Data to Knowledge (BD2K) analytics to detect multiple interdependent factors that result in ZIKV-related microcephaly [28].

## 2. Methods

A complex network of interdependent determinants of ZVI-microcephaly was generated by inferring the associations between variables describing diverse factors and the incidence of microcephaly attributed to ZVI in Brazilian municipalities during the first semester of 2016. This network provided the context to extract sub-networks of factors associated to ZVI^+^-microcephaly and controls that allowed us to perform comparative inferential analysis beyond a descriptive approach. Appendix A summarizes the process and analysis.

### 2.1. Data Collection, Integration, and Preprocessing

Secondary data were collected from Brazil‘s official websites about factors according to the WHO conceptual framework of the social determinants of health (SDH) [28] summarized in Table 1. Additionally, the data in Table 1 were included for ZVI-related microcephaly incidence from the open data initiative of the Brazilian system of public health surveillance and other health results.

By the standard identifier of the various Brazilian municipalities, we integrated an initial set of more than 700 variables for 5570 municipalities [29]. The pre-processing of these continuous, discrete, and binary variables included identifying and replacing missing values according to particular customary processes of collection and selecting n = 5565 complete cases (the population of municipalities). All the binary variables of the context determinant were included. Pairwise collinearity and concurvity detection assisted with the variable selection of the remaining numeric variables (651) of the structural and intermediary determinants and the health results. The collinearity criteria included Pearson correlations, their Fisher-z transform, and variance inflation factors (VIF) beyond 2, 2.5, and 3 standard deviations, and interquartile ranges (IQRs) of 2 and 3. The concurvity criteria were the Spearman correlations beyond 2 and 2.5 standard deviations and the IRQs. We also determined the distributions, thresholds, and matrices to visualize the different patterns of criteria that eased the team discussions about downsizing the number of variables to 382.

Using Cronbach’s alpha, we validated the construct reliability of the variables grouped in the four scales (context, structural, intermediary determinants, and health results) that arrayed 18 determinants. For each determinant, Table 1 describes the variables included, their source, their Cronbach’s alphas, and the colors assigned for further visualization of the results.

### 2.2. Network Generation

We obtained the general structure of the complex network of associations among the m = 382 final variables by inferring throughout the entirety of the data all possible linear Pearson and monotonic Spearman higher-order partial correlations for interval/ratio variables and equivalent point-biserial correlations for the binary variables (R package pcor) [30]. As no variable was previously labeled as dependent or independent, the undirected character of partial correlations fulfils the exploratory and objective scope of this study.

As higher-order partial correlations (pcor) control associations between each pair of variables was determined by removing the effects of the remaining full set of m-2 variables throughout the entirety of the data, we accounted for and reduced the confounding and spuriousness induced by any third variable. To quantitatively substantiate the global effect of confounding and the consequent reduction of spurious associations that resulted from using partial correlations, we also tested for differences between Fisher transformations of higher-order partial and zero-order coefficients.

Significant differences between the distributions of zero-order and partial correlation coefficients (*t*-test *p* < 0.001, 95% CI [0.1287–0.1224]) made explicit the presence of confounding throughout the data. A mean difference with a wide standard deviation of 0.12 ± 0.61 between both types of correlations endorsed our choice of higher-order partial correlations to infer the network structure.

Hartemink’s mutual information method (R package bnlearn) [31] was used to reconcile the different numbers of levels on the distributions of (m × (m − 1))/2 = 72,771 Pearson and Spearman partial correlations and their corresponding *p*-values (non-corrected and corrected by the family-wise error rate methods of Bonferroni and Holm–Šídák, and by the Benjamini–Hochberg false discovery rate procedure) to find the objective and meaningful thresholds. As all the strongest Spearman coefficients were included in lower and higher tail categories of the Pearson coefficients distribution, we maintained this last category to define the final seven thresholds and retain the meaningful intervals corresponding to the upper and lower two as the strongest positive (0.812, 1.000), strong positive (0.509, 0.812), strong negative (−0.711, −0.442], and strongest negative (−1.000, −0.711).

The resulting network obtained was composed of 99.73% of the initial variables as nodes and 36,400 statistically significant and meaningful associations as links. The customary indices in Appendix A summarize the overall properties of this general network structure. According to the KS test, with a large left tail, a negatively skewed degree distribution did not fit the normal, Poisson, or other functions, but an R^2^ = 0.394 showed a significant (but weak) fitting to a power law function. In this way, the dense general network was still asymmetric, allowing for some sort of differential organization.

As a network is formed by controlled associations corresponding to statistically based confounding-free relationships, different types of triads and related indices, such as Watts–Strogatz and clustering coefficients (CC1 and CC2), provide clues about interaction, which is relevant for epidemiological analysis. In this way, variables that are part of one edge triads shows pairwise associations independent of others, intransitive triads correspond to multiple independent associations, and transitive triads display synergistic or antagonistic interactions. In our network, transitive triads prevailed suggesting the possibility that the multiple factors associated with any outcome node might also be a modifier, in terms of effects.

Topological analysis required further reduction of the network density to look for an optimal consistent structure. By maximizing the difference between the clustering coefficients of the general network and a generated random Erdős–Rényi model, we identified a threshold of |0.76| absolute partial correlation. This thresholded network captures associations with the strongest transitive, and therefore interdependent, interactions. In addition, using ‘Graphlet Correlation’ distance, we used the 2 to 5 node graphlets [32], which corresponded to small subgraphs, to best capture the local wiring patterns (structures) around the nodes in the networks, and we measured the structural distance between two networks using their graphlet correlation distance [33] in which, to reduce the effect of structural decencies between graphlets, we only considered the 58 non-redundant 2 to 5 node graphlet orbits (GCD-58). We also computed the topological distance between the general network at various thresholds and a set of network models (Appendix A), and we found an additional threshold at |0.65| absolute partial correlation for the maximal distance to the Erdős–Rényi model. The resulting thresholded network captured more complex interactions among the groups of three to five factors and outcomes. The indices were also computed for the thresholded networks (Appendix A). 

For network visualization, by the energy-based Kamada–Kawai algorithm that included link weights, we obtained two-dimension coordinates in two levels among the determinant factors and among the nodes within factors. At the determinant level, the weights used were avg-pcor^2^, and at the node level, the weights were within-determinant pcor. The node colors were assigned previously, according to Appendix A.

We assessed the global significance and validity of the general network by statistical tests between these customary indices and the equivalent parameters of sampling distributions obtained by a Monte Carlo simulation (MCS) of 1000 random graphs, according to the Bernoulli variant of the Erdős–Rényi model based on the observed average degree [34]. Appendix A shows the confidence intervals of different indices as an estimate of the gap between the general network and random graphs. All network indices were significantly different from the corresponding parameters of the MCS-based sampling distribution, rejecting that the network structure obtained by higher-order partial correlations was due to chance. Additionally, the average of the Grapplet Degree Distribution (GDD) [35] of the general network to the Erdős–Rényi model was 0.66 ± 1.84 × 10^−5^.

The construct reliability of the determinants of health (the node groups in Appendix A), which was verified during data preprocessing by Cronbach’s alpha, was also confirmed by a canonical-correlation analysis that identified a statistically significant relationship between the determinants and graphlet orbits representing the local topology of the network at both the |0.76| and |0.65| thresholds [33]. Additionally, the canonical correlation analysis (in red) and the green plots display the complexity of the interactions among the determinant factors (Appendix A). According to the 1-cannonical variable from the network thresholded at 0.65, all pooled microcephaly surveillance incidence distributions were anticorrelated with ‘education’, and partially with ‘social networks’ and ‘biological factors’, and ‘occupation’ acted as a broker or mediator. Showing a closer interaction as part of the square structures, intermediate factors such as health system, material circumstances, other health results, and childhood development were predominantly closer to microcephaly surveillance. Income and social class were also close, but the first was anticorrelated. The two-cannonical variate shows that microcephaly surveillance, income, social networks, and macroeconomic policy belong to the interdependent square structures, while race/ethnicity, occupation, and the biological factors that act as mediators, as well as material circumstances, vector borne diseases, and health results, are more peripheral. As none of the canonical analysis with the network thresholded at 0.76 included microcephaly surveillance, the results showed how the remaining factors were interdependent and shaped the profiles of well-being in municipalities.

### 2.3. Inferential Network Analysis by Subnetwork Comparison

Testing the system of composite hypothesis relied on extracting four subnetworks from the full and thresholded (0.65 and 0.75) networks presented earlier. Subnetwork extraction relied on a k-neighborhood procedure which held the neighbor nodes closely associated with specific target nodes/variables. The target node of municipal distribution of 1810 cases of microcephaly with confirmed ZVI defined the m-ZVI^+^-subnetwork; the node of municipal distribution of 3924 microcephaly cases that were negative for ZVI defined the control m-ZVI^−^-subnetwork. The target variables of municipal distribution of 117,326 low birth weight (LBW) cases and 863,153 births (B) defined the additional control subnetworks. Figure 2 presents the indices describing each subnetwork.

By composite hypothesis testing using various statistical parametric and non-parametric methods to account for similarities and differences, we compared the components and topologies of subnetworks and their association patterns. The compared components included the vectors and distributions of neighbor nodes, unweighted and weighted links, and customary indices. The statistical tests included the coefficients and *p*-values of the zero-order Pearson, Spearman, Kendall, and Phi correlations, and the Student’s *t*, Wilcoxon signed rank, McNemar, and Kolmogorov–Smirnov (KS) values for corresponding independent and dependent samples [36]. We also identified the type of function that better described each subnetwork degree distribution by computing R^2^ determination coefficients with different reference models. For comparing the subnetwork topologies, we used Graphlet Correlation Distance (GCD), with GCD-58 already described.

To interpret the detailed meaning of the results of the composite hypothesis testing between the four subnetworks across the scales, determinants, and single variables, we focused on higher order squared partial correlations (pcor^2^) and their averages, which are the known measures of the effect size that account for the proportion of variance involved in each association without the effects of other variables [10]. For the scales and determinants, we computed averages (avg-pcor^2^) via Fisher’s z-transformation of pcor. However, the pcor and zero-order correlations (zcor) nuanced our findings that were mainly based on pcor^2^. In this way, pcor values similar to zcor values hinted at spurious associations, and pcor values higher than zcor values suggested the presence of suppressor variables [36,37].

Additionally, by color bands that were extracted from the corresponding heat map matrices of zcor, pcor, and pcor^2^ for the full and thresholded subnetworks, we complemented the analysis with the visual profiles, or fingerprints, of the direct connections for each subnetwork (Appendix A). In pcor^2^ bands, the darker blue shade designates high link values, while the lighter shades indicate low link values. The fingerprint bands for zcor are represented by a continuous ascendant spectrum between red, yellow, and green, while the colors of the pcor fingerprint bands follow the same four intervals for positive and negative strong and strongest links, as shown in the network visualizations.

In the Appendix A, we present the additional data used to collect our data, in accordance with the open policy initiative related to the Zika Virus Disease and an analysis of the network by testing the composite hypotheses using BD2K advanced network science. See Table 1 for the detailed sources of the available data that were utilized.

## 3. Results

If the Zika infections explained the microcephaly clusters, then the observed distribution of Zika should have been at least roughly proportional to the observed distribution of microcephaly cases. Since it was not, it seems reasonable to hypothesize that the observed non-proportional distribution might be due to hidden and/or multiple factors. Further, these factors might be shared with other concurrent health phenomena. To test the hypothesis concerning multiple factors related to microcephaly incidence positive for Zika infection (m-ZIKV^+^), we explored its specific pattern of associations with multiple factors, and we contrasted that pattern with the patterns of associations with these factors that were exhibited in controls of three types: (1) microcephaly incidence negative for Zika infection (m-ZIKV^−^), (2) low birth weight (LBW, an acknowledged indicator of infant morbidity), and (3) births (B, as a proxy for mainly healthy newborns), as seen in Figure 2.

In the first step of modeling, the public frequency data was collected from multiple official sources to calculate all of the controlled statistical associations among the 382 variables describing the 16 determinant factors, the incidence of concurrent health outcomes, and m-ZIKV^+^ surveillance in 5565 Brazilian municipalities. All the variables and determinants are described above in Materials and Methods or in the Appendix A. To control the associations from spuriousness and confounding, we used squared higher-order partial correlations (pcor^2^) as proxies for effect size, complemented by zero-order correlations (zcor) and partial correlations (pcor), each with the corresponding significant *p*-values evaluated after correction for multiple comparisons. We identified and kept the strongest associations according to thresholds of the full distribution of pcor weights and, by graphlet, the optimal distance from an Erdős–Rényi random network that highlighted the links with maximum structural relevance. In a context network, all of the significant controlled and strongest associations were mapped as links between nodes representing the variables (Figure 3). Additionally, the network was validated by comparing it with a null equivalent Erdős–Rényi network to test for randomness.

In the second step of analysis, we described and evaluated the patterns of factors associated with case and control phenomena by extracting the corresponding subnetworks. The case subnetwork included all the interdependent nodes directly linked to m-ZIKV^+^ incidence (Figure 2, Panel A). The control subnetworks included all the nodes directly linked with m-ZIKV^−^ incidence (Figure 2, Panel B), LBW (Figure 2, Panel C), and BIR (Figure 2, Panel D). The macroanalysis compared the subnetwork structures using statistical tests between the distributions of the network indices and by using advanced network structural methods of graphlet distance. A microanalysis assessed the details explaining the differences by comparing the fingerprints of all the aligned subnetworks. These subnetwork fingerprints corresponded to the visualizations of ordered weighted associations between the matching pairs of variables in each subnetwork (described in Appendix A and Materials and Methods).

### 3.1. General Pattern of Factors Associated with Microcephaly Is Not Specific for m-ZIKV^+^

In general agreement with the recent work reported by Brady et al., (2019) [2] the comparisons by different methods suggested that the patterns of factors associated with the case m-ZIKV^+^ and control m-ZIKV^−^ subnetworks were analogous. In the present study, the subnetwork size and indices were very similar (average degrees were slightly different, and only the differences between each subnetwork closeness and betweenness vectors were significant) and nearly all parametric and nonparametric statistical tests supported the nonsignificant differences between the nodes present and distributions of pcor and pcor^2^ link weights in each subnetwork (Table 1). According to the R^2^ determination coefficients, the degree distributions of both subnetworks fit the same functions (cubic, quadratic, and linear) and were graphically close (Figure 2, Panel A)**,** and they had a significantly similar percentile-based KS test. By *t*-test, the vectors of weighted degree and clustering coefficients were non-significantly different. The GCD-58 between both subnetworks was the smallest among all the comparisons.

These results illuminated latent factors and their central roles and convergence to suggest that effects related to the ZIKV^+^-microcephaly incidence in Brazil were not specific to Zika infection and were common to both types of microcephaly. A diagnostic failure to discern Zika infection during the epidemic might explain this result. This possibility suggests an underestimation of the size of Brazil’s ZIKV^+^ microcephaly incidence. Alternatively, assuming that diagnostic uncertainty similarly affected both groups, it is evident that microcephaly is a complex event where multiple causes and mediators converge. As Schwartz [38] suggests, Zika diagnostic tests deserve a general inclusion in antenatal screenings, even more so if multiple modes of transmission increase the uncertainty about infection risk. Above all, improving diagnostic tests is critical [39].

The comparisons of m-ZIKV^+^ with the LBW and BIR control subnetworks validate the former results. As expected, we found a partial similarity between the association patterns of the m-ZIKV^+^ and LBW subnetworks because they are different events, but they affect common childhood populations. Except for the McNemara test, all the other statistical tests showed dependence and high correlations between the nodes in both subnetworks. Despite significant differences between the means and medians of pcor link weights, the data were highly correlated. Conversely, the mean and median effects of the factors by pcor^2^ were insignificantly different and correlated, but less than for m-ZIKV^+^ and m-ZIKV^−^, which is evident upon a visual comparison of the link weights (Figure 4, Panel B). The degree distributions were also similar with the non-linear functions and smaller R^2^s than in the comparisons of the m-ZIKV^+^ and m-ZIKV^−^ subnetworks, and the GCD-58 distances were observed to be higher (Table 1).

### 3.2. Factors That Explained the Non-Specific Pattern of Associations of m-ZIKV^+^

Conversely, but not surprisingly, the comparisons between the m-ZIKV^+^ and BIR subnetworks largely revealed differences. Except for the insignificantly different average and median effects of the factors by pcor^2^, almost all the tests showed significant differences and low correlations for the nodes and link weights between the m-ZIKV^+^ and B subnetworks (Table 1). A visual comparison shows (Figure 4C) divergent pcor^2^ and pcor sign patterns. The two GCD-58 values were the highest among the comparisons of the m-ZIKV^+^-subnetwork with each of the three control subnetworks (Table 1). These results are consistent with the differences between ill and healthy children.

Further detailing the factors that shaped m-ZIKV^+^ and simultaneously addressing the similarities and contrasts with control subnetworks allows for a more accurate assessment of the role of each factor. For the m-ZIKV^+^-subnetwork, the avg-pcor^2^ correlations showed weak average effects of the factors we classified into the WHO’s three hierarchical scales (the socioeconomic and political context, social structure, and intermediary factors). Among these, the intermediary factors had a slightly higher influence. By contrast, for the LBW subnetwork, factors in the social structure and intermediary scales were salient. For the B subnetwork, factors in all three scales had low-to-moderate effects. All values of the corresponding coefficients for the scales are shown in Figure 4A and for the factors in Figure 4B.

The factors in the scale of the socioeconomic-political context (Table 1 and Figure 4B) involve broad aspects of the social system policy choices and welfare state programs such as political ideologies, economic regulations, governmental interventions in the economy, environmental regulations, and educational systems, as well as population demographic patterns such as growth rate, age distribution, and career attitudes. Although these factors generally “elude quantification”, we used the availability of municipal policies as proxies for the awareness of and concern about the diverse issues in the socioeconomic-political context [27]. For the factors belonging to the political context scale, the avg-pcor and avg-pcor^2^ gave evidence of only a few weak effects on the m-ZIKV^+^ subnetwork. The factors with these effects were the policies about ‘culture and social values’, ‘environment’, and ‘demographic conditions’, the latter linking with all population age groups, particularly with aged, adult, and child populations.

There were other factors in the socioeconomic-political context scale that had almost identical patterns of association with the m-ZIKV^+^ and m-ZIKV^—^ subnetworks. However, links with the nodes of Gini and Theil indices defined a slight influence of ‘macroeconomy’ on the latter. In the control LBW subnetwork, the avg-pcor^2^ was lower than in the m-ZIKV^+^ subnetwork due to a balance of weaker avg-pcor^2^s of demographics, culture, governance, and macroeconomic factors. In the control B subnetwork, the culture factor had the highest avg-pcor^2^ (with an opposite sign of pcor), followed by the factors of existing social and public policies. The avg-pcor^2^ of demographics was lower in the B subnetwork than in the m-ZIKV^+^ subnetwork, and almost all age groups had a moderate to low pcor^2^. These results showed that policy context profiles and concerns mainly impact long-term problems. In the case of emergent problems, such as the microcephaly clusters linked to ZIKV^+^**,** associations with policy were restricted to environment, sanitation, water, culture, and education, while a broader set of factors was present in LBW and B. Despite the fact that the demographic context is present in all networks, the opposite associations defined normal and pathological patterns. Predominantly adult populations in municipalities were shown to be specifically related with microcephaly.

The factors in the scale of the structural determinants (Table 1 and Figure 4B) include socioeconomic processes shaping the unequal social position of individuals [25]. In the m-ZIKV^+^ subnetwork, links with race/ethnicity and social class had the highest avg-pcor^2^. All races were associated, but the municipal proportions of white and yellow races had a proportional pcor with the m-ZIKV^+^ node. The factor of social class mainly included variables describing vulnerability. Additionally, the ‘occupation’ factor showed a moderate avg-pcor^2^ by the nodes of ‘labor force’, ‘unemployment proportion’ (general and by race), and ‘unemployment in the environmental field’ as the only specific occupational sectors involved. Conversely, the ‘income’ and ‘education’ factors had very low avg-pcor^2^ values. Nevertheless, inside the income factor, ‘mean income’ and ‘Theil index for work revenues’ had high pcor^2^s, while ‘income of 20% richest’ and ‘per-capita GDP’ had moderate pcor^2^s. Inside the education factor, only ‘enrollment of 5-6-year-old’ and ‘illiteracy’ were associated with, respectively, moderate and low pcor^2^s. Minor differences in the m-ZIKV^−^ subnetwork were due to links with ‘unemployment in the construction sector’ inside the occupation factor and link absence of ‘per-capita GDP’ in the income factor (Figure 4, Panel B). In this way, ZIKV^+^ and ZIKV^−^ microcephaly have effects of the macroeconomic tendency of unequal income distribution, which is explained in the social structural scale by the potential and blocked access to income of those who, respectively, belong to the labor force or are unemployed, and this seemed to operate in different ways among the races, with more effects in municipalities with larger white and yellow populations.

In the control LBW subnetwork, the determinants of race, social vulnerability, and occupation in the structural scale stood out based on their avg-pcor^2^ values. Within the race determinant, white, mixed, and black had the highest pcor^2^s and all social vulnerability variables were linked, except ‘woman head of family’. These association patterns inside each of the determinants were similar to those in the m-ZIKV^+^ subnetwork. In the control B subnetwork, the determinant of race also had the highest avg-pcor^2^, but, in contrast to the associations in the m-ZIKV^+^ subnetwork, education, income, and occupation were relevant. Per capita GDP, illiteracy (determined without reference to race), primary school enrollment, participation in the labor force by members of yellow and indigenous races, and self-employment were remarkable nodes with high pcor^2^ values. While the LBW subnetwork and the B subnetwork had significant associations with a number of the same structural determinants, the telling fact was that those associations had opposite signs for each subnetwork, respectively. The patterns of vulnerability, low education, income, and racial inequalities seemed to be the strongest candidate factors for further action on microcephaly in Brazil.

The intermediary scale factors (Table 1 and Figure 4) convey the underlying social stratification of the structural factors by determining the differences in exposure and vulnerability to health-compromising conditions [25]. The intermediary scale includes material, psychosocial, behavioral, and biological factors. Among the three scales, the intermediary scale contributed the most to the m-ZIKV^+^ subnetwork (Figure 3), and all its factors were associated with that subnetwork. ‘Childhood development’ had the highest avg-pcor^2^ of the factors, mainly due to the value of this factor in the LBW subnetwork. Similarly, the avg-pcor^2^ of the intermediary scale in the LBW subnetwork was the highest due to the contributions of all determinants, with its main roles in the ‘undernourishment’ node and factors of ‘social relationships’ and the ‘health system’. In the B subnetwork, conversely, but as expected, association with the problem of ‘childhood development’ was absent.

The factor of social relationships includes variables about marital status. Most marital status variables had associations in the m-ZIKV^+^ subnetwork. The variable ‘single’ and the variable ‘never couple living’ both had negative pcors for playing a protective role with respect to m-ZIKV^+^. The former was linked by a high pcor^2^ to the subnetwork and the latter, by a moderate pcor^2^. These variables had similar connections to the control LBW subnetwork, but in the B subnetwork, the connections were weaker and had opposite signs. Thus, the data support the view that different types of family relationships (protective/stable, unstable, and weak) offer a similar substrate to cases of microcephaly and LBW, but not to cases of normal birth. However, the m-ZIKV^+^ subnetwork had no connections with the vulnerable characteristics of households that were present, with negative effects, in the LBW and B subnetworks.

As anticipated for the biological factors, the m-ZIKV^+^ subnetwork had associations with the pregnancy, fertile age, and newborn and early childhood populations. However, according to the different signs of pcor, links with the female population above 40 years old could act as a modifying factor for Zika infection. These variables were also present in the LBW subnetwork but were absent in the B subnetwork. Unlike the m-ZIKV^+^ and m-ZIKV^−^ subnetworks, the control LBW and B subnetworks had other connections with nodes representing demographic indicators such as ‘life expectancy at birth’, ‘human development index longevity dimension’, ‘late-age survival probabilities’, and ‘total fertility rate’. The reasons for this difference are the sustained effects LBW and births have on population dynamics. Because of the acute character of the microcephaly clusters, we found no effect on demographic indicators in the epidemic.

The role of the health system in health outcomes involves access to care and health interventions. We refer to these factors as health outcome determinants (Table 1 and Figure 4). This factor had a similar avg-pcor^2^ in the case and control subnetworks, but the opposite sign of the avg-pcor in the B subnetwork comports with the health care factor having a different role for healthy and ill populations. Immunizations are recognized preventive health measures and are the focus of action where infectious diseases threaten populations. However, immunity interventions are not straightforward and encompass diverse results from the positive (e.g., cross-immunity) to the negative (e.g., sporadic secondary undesired effects). Associations of m-ZIKV^+^ with doses of multiple customary vaccinations during 2015 and 2016 had highpcor^2^s, including those targeting females and pregnant women. However, by means of pcor^2^, we did find a precedent strong association with 2015 ‘influenza doses for pregnant women’ and a slight association with 2015 ‘triple viral for fertile-age women’. For 2016 vaccinations, ‘influenza doses for pregnant women’ had the highest pcor^2^. We also found associations with doses of ‘yellow fever vaccine’. In almost all vaccination doses and coverage, a negative pcor prevailed, with important sign shifts from zcor to hint at overall protective roles, which can be confounded by suppressor variables.

In the control LBW subnetwork, a wider variety of customary vaccinations during 2015 and 2016 were associated with a rather different pattern (Figure 4C), even though vaccine doses during 2015 of ‘influenza for pregnant women’ and ‘triple viral for fertile-age women’, and vaccine doses during 2015–2016 of ‘yellow fever’ were also present with lower pcor^2^s. An even broader pattern of vaccinations was present in the B subnetwork, with almost all pcors with opposite signs. This allows for the grounding interpretations of other subnetworks and for noticing the potential protective role of vaccinations.

Among other health care interventions, the connection between ‘prenatal care’ and m-ZIKV^+^ was the strongest, possibly due to increases in health care once developmental problems in the fetus are identified [40]. For this factor, pcor^2^ values successively decreased in the m-ZIKV^−^ subnetwork to the LBW subnetwork and then to the B subnetwork. Additionally, the opposite signs for pcor gave evidence that the role of prenatal care is different for ill and healthy populations. Data analysis also showed differences between general health funding and local health funding. While ‘total health funds’ had a similar pattern of associations across the subnetworks, except that the B subnetwork had the highest coefficient, local health funding from municipalities was correlated with ZIKV^+^ and ZIKV^+^-microcephaly. The difference shows that general health funds target long-term issues and local funds address emerging problems. In the same way, the different remaining indicators of health care access and interventions were connected only in the LBW and B subnetworks.

Material circumstances that shape the local environment, where vectors, infectious agents, and people converge, have a central role in VBD. Despite the weak avg-pcor^2^ of this factor overall, in the m-ZIKV^+^ subnetwork we found strong contributions of single nodes of the factors that represented mosquito populations, public services, and dwelling materials. However, more general variables about settlement, land use, and agrochemicals management were not connected to the m-ZIKV^+^ subnetwork. Entomological surveillance addressed the vector populations by an index for the rapid assessment of Aedes larvae (LIRA). Despite the fact that the total LIRA was not linked to the m-ZIKV^+^ subnetwork, in that subnetwork, we found a high pcor^2^ for LIRA in water, residence, and waste. In the LBW subnetwork, LIRA in water also had a high pcor^2^ but LIRA in residence and waste was low. In the B subnetwork, LIRA in waste and residence also had salient connections, but LIRA in water was not associated. The presence of strong associations of LIRA in the case and control subnetworks is evidence of a generalized presence of vectors in the country. However, for the illness-related subnetworks of m-ZIKV^+^, m-ZIKV^−^, and LBW, the predominance of LIRA in water denotes conditions that are relevant for the reproduction of mosquitoes, while sign shifts between pcor and zcor signal the involvement of other variables. Additionally, the common LIRA patterns in the m-ZIKV^+^ and m-ZIKV^−^ subnetworks are consistent with the special vertical route of transmission from mother to child that indirectly depends on precedent vector and sexual transmission routes.

Despite the significant association of the presence of LIRA in water, residence, and waste with the m-ZIKV^+^ subnetwork, the association of suboptimal access to water and sanitation within this subnetwork was weak. It was similarly counterintuitive that both standard and substandard dwelling materials were linked to this subnetwork. In the LBW subnetwork, there were links with high pcor^2^s to even more types of dwellings. Associations with a broader spectrum of conditions were found in the B subnetwork. Together, the LIRA and dwelling results provide evidence that proper conditions for mosquito breeding are mainly due to how people handle water and not to structural conditions.

Our study of associations of Zika infection with other concurrent health outcomes revealed both synergistic and antagonistic interactions with other diseases. Such interactions moderate the characteristic features of Zika infection. For example, not only do some VBDs, such as dengue, share vectors with Zika, some studies are focusing on possible cross-immunity. In the m-ZIKV^+^ subnetwork, we found strong associations with the municipal incidence of malaria and Chagas diseases occurring among children and moderate associations with malaria occurring in a municipality’s remaining age groups. In the LBW subnetwork, there were similar associations with municipal incidences of malaria and Chagas diseases. However, as we found that malaria and Chagas disease in childhood populations had weaker associations with the opposite sign in the B subnetwork, their role in the m-ZIKV^+^ subnetwork seems antagonistic. We found nonsignificant associations with the incidence of severe dengue in the case and control subnetworks. Associations with the incidence of congenital syphilis and infant and child mortality were absent in the m-ZIKV^+^ subnetwork but present in the LBW and B subnetworks.

The incidence of poisoning by certain toxicants and toxins provided proxy information about exposures in municipalities. Such exposures have been linked to noninfectious microcephaly, but they also emerged as alternative explanations of the Zika-related Brazil clusters. During 2015, the total poisonings by municipality of notification, residence, and exposure had high pcor^2^s within the m-ZIKV^+^ subnetwork. In that subnetwork, poisoning due to exposure to metals had the highest pcor^2^, followed by poisoning due to exposure to agrochemicals used in public health. A similar pattern appeared in the m-ZIKV^−^ subnetwork. By comparison, in the LBW subnetwork, poisoning due to exposure to agrochemicals used in public health had an even higher pcor^2^ and poisoning due to exposure to metals had a lower pcor^2^. In the B subnetwork, poisoning due to exposure to metals had the lowest correlation and links with poisoning due to exposure to agrochemicals used in public health were absent. Although exposure to toxic metals seems to be generalized (being found in all subnetworks), the strong associations we found were with both ZIKV^+^ and ZIKV^−^ microcephaly. Conversely, exposure to agrochemicals used in public health was stronger in LBW and not correlated with healthy births. Our results suggest the role of agrochemicals used in public health and find a new pattern with exposure to metals and other agrochemicals [41,42] possibly contributing to microcephaly outcomes related to Zika infection during pregnancy.

## 4. Discussion

By exploring the interdependencies among several variables from the available official public data, we have advanced the understanding of the additional factors contributing to Zika-related microcephaly. As complex interdependencies go beyond mere pairwise associations between each factor and m-Zika incidence, using methods that address effects of third variables, we reduced spuriousness to discover depurated relationships among a full set of 382 variables and mapped them in a context network (Figure 3). To test whether ZIKV^+^-microcephaly was a fully differentiable event, we extracted one case and three control subnetworks that enclosed specific patterns of associations with m-ZIKV^+^, m-ZIKV^−^, LBW, and B, respectively, to be compared on a macroscale by the inferential statistics of the network parameters and structural measures. Additionally, we comparatively explored the microscale of the subnetworks to explain the differences and similarities among the case and control subnetworks.

Our network included the municipal distributions of 1810 cases of m-ZIKV^+^ and 3924 cases of m-ZIKV^−^ during 2016. Previously, the reported incidence of microcephaly in Brazil was fewer than 200 cases per year. That the number of ZIKV^−^ cases in 2016 was more than double the previously reported number of ZIKV^+^ cases was the greatest source of uncertainty in this enormous, unexplained rise in microcephaly. Through our inferential approach, which was strongly framed on the control subnetworks of LBW and B, we disclosed candidate factors to use in generating both hypotheses about the modulating effects on m-ZIKV^+^ and the alternative explanations for m-ZIKV^−^.

Brazil’s demography, macroeconomy, established health problems such as LBW, and policies set the context for the incoming microcephaly clusters. Certain policies appear to have had an effect on the respective m-ZIKV^+^ and m-ZIKV^−^ clusters. These were specific environmental policies and policies showing an awareness about municipal cultural backgrounds and behaviors. However, the effects of the intended interventions under other policies were poorly incorporated into the respective clusters. We also observed that, in salient contrast to the m-ZIKV^+^ clusters, the m-ZIKV^−^ and LBW subnetworks were negatively related to macroeconomic indices.

The summary analysis (Figure 5) discerned a few intricate associations of the m-ZIKV^+^ clusters with the complex socioeconomic circumstances of access to income, race, and education, which are closer to individuals than to macroeconomic indices, and they frame unequal social positions. These associations were fewer than those that were related in similar ways to the incidence of LBW and BIR. Further, we found no associations between m-ZIKV^+^ incidence and variables that identified the incidence of vulnerable families in municipalities. A household gathers together the socioeconomic status of each of its individual members into a combination that can be complex and diverse. Yet, we saw that households of diverse types of couples heading families were similarly linked to the m-ZIKV^+^ and LBW subnetworks.

Intermediary factors had the predominant effects on m-ZIKV^+^ municipal incidence. Inside that scale, the factor of the population of fertile-age females beyond the age of 40 played a role in the m-ZIKV^+^, m-ZIKV^−^, and LBW subnetworks, where the mothers were more likely to be in this age group. Despite the fact that vectors seemed to be ubiquitous in Brazil, according to the LIRA indices present in the case and control subnetworks, both coefficients and the lack of links with the incidence of severe dengue (another VBD with the same vector) revealed that circumstances easing the growth of mosquito populations had an indirect role coherent with the vertical path of Zika transmission causing microcephaly. Additionally, the associations of m-ZIKV^+^ with access to aqueducts and waste collection point to the inadequate management of water as the explanation for the existence of conditions conducive to mosquito breeding.

The recently described public health exposome (PHE) framework with Big Data to Knowledge (BD2K) analytics is a novel environmental public health framework that could assist local urban public health and rural municipalities in a post-epidemic evaluation of Zika infections in Brazil. The framework could be used to predict and model the relationships between environmental and socio-demographic variables and adverse health outcomes [28]. In this instance and going forward, the PHE would be available to assist public health officials in northeast Brazil with integrating, partitioning, and clarifying pockets of increasing prevalence in low birth weight and pre-term births from LBW subnetworks containing both m-ZIKV^+^ and m-ZIKV^−^. Additionally, the PHE framework would also inform the question as to how chemical stressors such as components of ambient pollution negatively impact and potentiate adverse pregnancy outcomes. We now know that there is sufficient evidence in 2020 to support the fact that exposure to chemical stressors causes important adverse health outcomes that result in premature death, including ischemic cardiovascular disease, stroke, respiratory infections, chronic obstructive pulmonary disease, and lung cancer [39,43]. By using the public health exposome [26] and BD2K analytics to integrate environmental science with population-based study designs, there is potential to contribute novel and innovative theories and interventions to dampen negative outcomes by assessing the cumulative risk from exposures to chemical and non-chemical stressors.

Public heath interventions include the spread of agrochemicals to control mosquito populations. An extensive presence of mosquitoes implies widespread exposure to these agrochemicals. Therefore, in addition to the links intuitively found between Zika infection, its vector, and m-ZIKV^+^, a link between the co-occurrence of this nonselective exposure and microcephaly is suggested. These data are in support of molecular data on exposures to the insecticide pyriproxyfen with a dampening of neurodevelopmental processes, particularly in areas with ZIKV prevalence [44]. Using methods that control for confounders, we found that poisonings due to these agrochemicals as proxies of exposure had the lowest associations with m-ZIKV^+^. An opposite pattern revealed co-exposure to metals as a potential factor contributing to m-ZIKV^+^. Exposure to metals has the potential to partially explain the m-ZIKV^−^ outbreak.

Lastly, we confirmed that several vaccinations are widely available as public health interventions, with a protective nonspecific role in m-ZIKV^+^. Understandably, health care factors were associated with all the case and control subnetworks. However, local health funding from municipalities and basic performance measures were pre-eminent in the m-ZIKV^+^ subnetwork. We observed that general health funding focused on local and long-term issues, as well as on emerging problems. However, the birth of 5734 microcephalic babies in one year produced lasting manifestations of both the Zika virus epidemic and the microcephaly outbreak that require the further refinement of health care resource allocation toward both public health preventive interventions and continuing care.

## 5. Conclusions

Further study is needed to test the ecological model and that the use of secondary data has limitations. The unique capability of using the *Public Health Exposome* framework and analytics is to allow the investigation into databases to examine variables and factors contributing to the risk trajectory of disparate health outcomes. The utility of our framework is demonstrated in the present study where multicausality of microcephaly beyond Zika infection was demonstrated [45]. Furthermore, while we agree that further studies are needed, primary data collection typically obtained through comprehensive environmental public health studies, epidemiological cohort or case control studies is often difficult to carry out because of the limited public health and research infrastructure in remote areas with the highest number of Zika cases [46,47,48]. Further, these studies would be very expensive and the findings would be delayed, which would postpone time-sensitive mitigation strategies, interventions, and/or control measures. Such interventions are particularly time-sensitive to mitigate against neurotoxicant exposures. 

## Figures and Tables

**Figure 1 ijerph-19-09051-f001:**
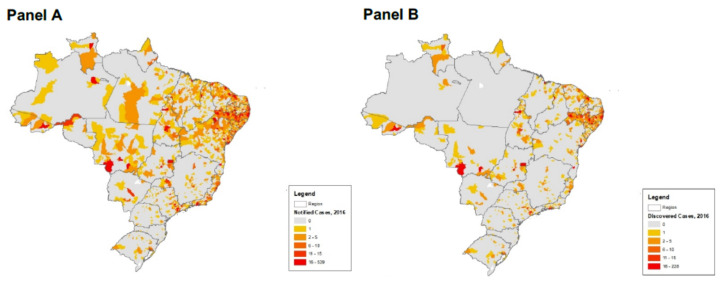
Geographic distribution of ZIKV incidence (Panel **A**) did not match with ZIKV microencephaly cases (Panel **B**) in the 2015-2016 ZIKV epidemic. See text for details. (Data courtesy of the Health Disparities Research Institute Mapping Center at Meharry Medical College).

**Figure 2 ijerph-19-09051-f002:**
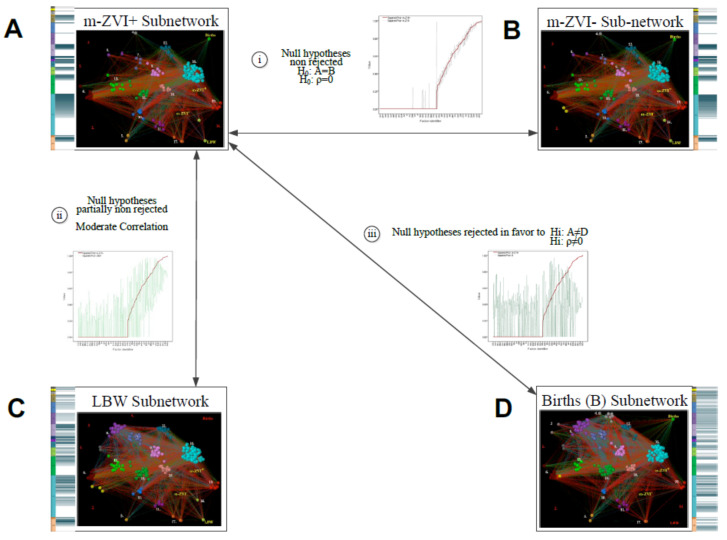
Summary of Statistical Comparisons Between Subnetworks. Boxes show visualizations of each subnetwork structure and fingerprint (**A**. m-ZIKV^+^, **B**. m-ZIKV^−^, **C**. LBW, **D**. B). Visualizations include colored nodes according to determinants described in Table 1, and links with colors according to pcor association strength (blue: strongest positive, green: strong positive, red: strong negative, dark red: strongest negative). Fingerprints in different shades of teal correspond to pcor^2^ values of direct links with each node/variable and case and control nodes. (Darker color bands correspond to stronger links, lighter correspond to weaker links direct links pcor^2^ and white corresponds to link absence). Arrows show comparisons between subnetworks: i; comparison between m-ZIKV^+^ and m-ZIKV^−^ subnetworks showed no significant differences of variables, links and weights of links, and solely significant differences of betweenness; ii; comparison between m-ZIKV^+^ and LBW had a partial concordance of networks (0.38), means and medians difference of pcor with similar sizes, and moderate to low concordance of effects (pcor^2^); and iii; Comparison between m-ZIKV^+^ and B, different for nodes, links and their pcor weights, but some similar effects by pcor^2^. Each line has overlapped distributions according to the ranked pcor^2^ of m-ZIKV^+^ (red line) showing the differences with the corresponding pcor^2^ of links of aligned nodes in m-ZIKV^−^ (i), LBW (green line in ii) and B (dark green line in iii) subnetworks. Oscillation amplitude of control subnetworks pcor^2^ show the smallest differences between m-ZIKV^+^ and m-ZIKV^−^, moderate differences between m-ZIKV^+^ and LBW, and the biggest differences between m-ZIKV^+^ and B. Flat red line indicates no connections.

**Figure 3 ijerph-19-09051-f003:**
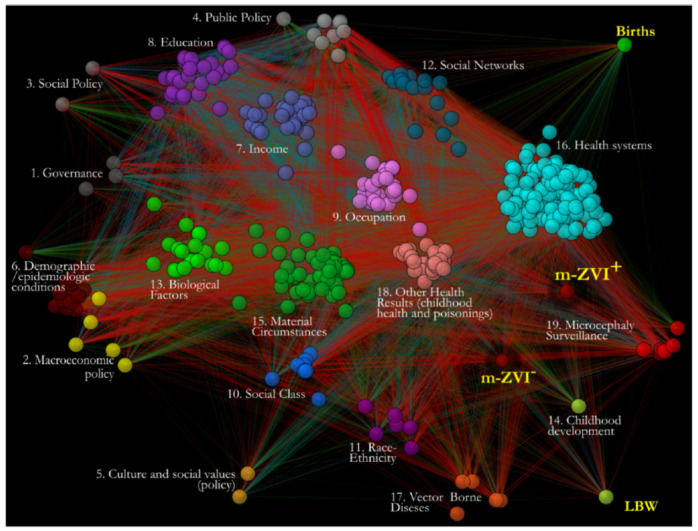
Context networks derived from Public Health *Exposome* framework and BD2K analytics demonstrates associations of multiple determinant factors with Zika Virus Disease incidence in Brazil’s municipalities during the 2015–2016 outbreak. The full network is shown where four colors show strongest positive partial correlations (dark blue), strong (green), negative strongest (dark red), and strong (light red) partial correlations among variables/nodes. Visualizations include colored nodes according to determinants described in Table 1, and links with colors according to pcor association strength (blue: strongest positive, green: strong positive, red: strong negative, dark red: strongest negative). Fingerprints in different shades of teal correspond to pcor^2^ values of direct links with each node/variable and case and control nodes. (darker color bands correspond to stronger links, lighter correspond to weaker links direct links pcor^2^ and white corresponds to link absence). Arrows show comparisons between subnetworks as explained in the legend for Figure 2.

**Figure 4 ijerph-19-09051-f004:**
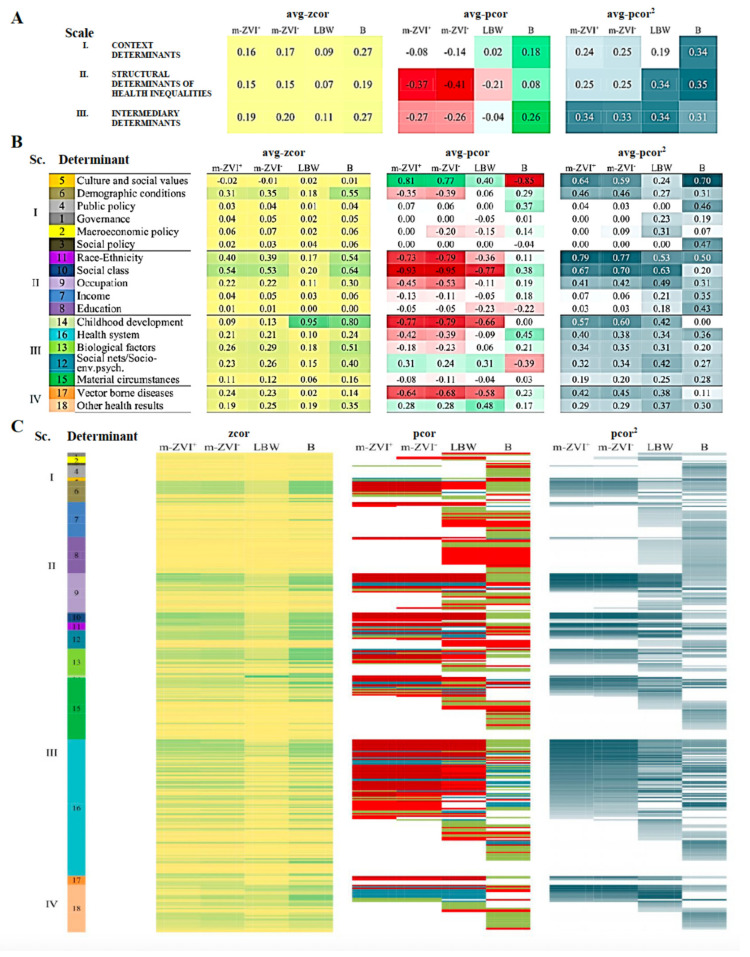
General pattern of factors associated microencephaly is unspecific for microcephaly incidence positive for Zika infection (m-ZIKV^+^) and these factors explain the nonspecific pattern of associations of m-ZIKV^+^. See text for details.

**Figure 5 ijerph-19-09051-f005:**
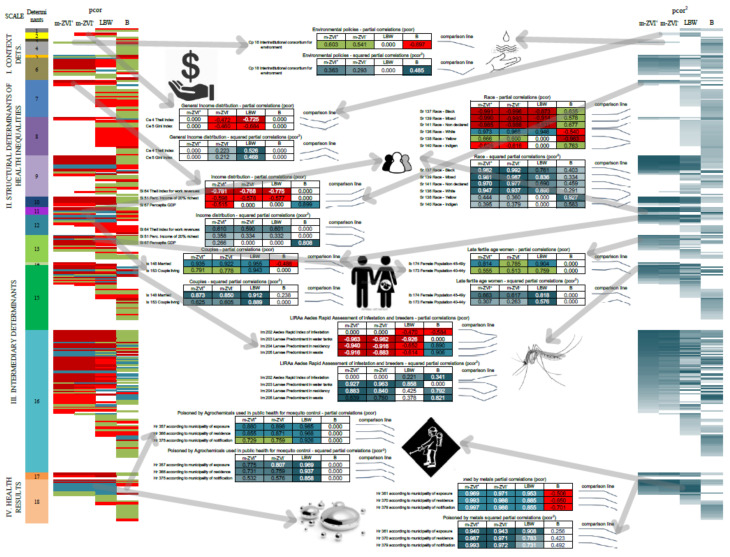
Summary of the Analysis. The right and left margins show case and control subnetworks comparative fingerprints of pcor and pcor^2^ values by ordered color gradients. These fingerprints are analogous to single columns of heat maps. The descending order of pcor^2^ inside each determinant of the m-ZIKV^+^ subnetwork defines the order of color gradients for all other control networks. Between fingerprints at margins, we extracted (according to arrows) the values that detailed each compared pcor and pcor^2^ of subnetworks. From top to bottom, environmental policies common to both types of microcephaly. Next Inward, a table shows differences between macroeconomic indicators between both types of microcephaly followed by income municipal unequal distribution, and inversely proportional relationship of both types of microcephaly with higher income populations. Differential involvement of white, couples and late fertile age women populations showed potential biological vulnerabilities. The relevant relationships of ZIKV-related microcephaly with mosquito breeding but not with mosquito populations other than non-specific relationships with agrochemicals used in public health are also present and are even higher with LBW. This would tend to dampen the effect and the effectiveness of this intervention. The last relationship and higher values for both types of microcephaly is an emergent factor to explain non-ZIKV-related microcephaly which may be an indirect moderator of ZIKV-related microcephaly.

**Table 1 ijerph-19-09051-t001:** Determinants of health and health outcomes related to microcephaly attributed to ZVI included in the model. For each determinant (arrows = Context, Structural, Intermediary, and Health) are included the Cronbach alpha values. [0.70 and above is good, 0.80 and above is better, and 0.90 and above is best].

Scale	Determinant		Summary of Variables in Each Determinant
Context determinants		1. Governance (0.434)	Governance was approached as the formalized convergence of diverse social actors of decision making, such as municipal planning agencies, mechanisms for empowering citizens and involvement in high level policy agendas [12].
		2. Macroeconomic policy (0.361)	Variables describe the balance through economic impositions and fostering, and the resulting economic inequalities in population by Gini and Theil indices [2,12,13,14].
		3. Social policy (0.149)	Variables about social support in municipalities through proxies of ZEIS (Zonas Especiais de Interesse Social) for low income housing and interinstitutional social support and development [2,12,13,14].
		4. Public policy (0.786)	Variables describe diverse instruments such as plans and legislations that stabilize general public policies on environment, land use, urban settlement, development and housing, economy and transportation.
		5. Culture and social values (0.872)	Variables about institutional concerns of cultural and educational affairs in municipalities [12].
		6. Demographic Epidemiologic conditions (0.598)	Population age pyramid according to the last 2010 census and 2015 population estimate for municipalities. Institutional concern of health and sanitation in municipalities [12,15].
Structural determinants		7. Income (0.755)	Population percent and income distribution between rich, vulnerable, poor and poorest populations, and for workers and unemployed. We also included the mean income by race, and the GDP, per capita GDP and the human development index (HDI) and its income dimension in municipalities [14,16].
		8. Education (0.878)	Attainment and enrollment in basic school (primary and middle), literacy by race, sex and according to age thresholds, and the expected years of study. We included the dimension of ‘education’ of HDI [14].
		9. Occupation (0.876)	Participation in the work force and unemployment, differentiated by race, educational attainment and work sector, and commuting [12,14,16].
		10. Social Class (0.976)	Social class was negatively approached by individuals and households located in agglomerates qualified as subnormal, and variables labeling social vulnerability [13].
		11. Race-Ethnicity (0.926)	Distribution by Brazil’s racial groups [2,12,13].
Intermediary determinants		12. Social Networks/Socio-env. Psych. Circumstances (0.896)	Civil status, people and children in households supported by people without education of who are vulnerable, and dependency ratio [12,14,16].
		13. Biological Factors (0.993)	Population distributed by sex, and women population in fertile age groups, fertility rate, infant female and male populations, life expectancy, longevity and ageing rate. Longevity dimension of the HDI [16].
		14. Childhood Development (0.948)	Negative approach by Low Birth Weight (LBW) and less than 1-year undernourished children [12,16].
		15. Material Circumstances (0.885)	Dwelling material, availability and access to aqueduct/in-house pipe water, sewage system, and waste disposal. Access to electricity. Vegetation distribution and land use. Potential exposure to agro-toxic by agricultural use or residue disposal. Rapid assessment of indices for *Aedes aegypti* (*Levantamiento Rápido de Indices para Aedes aegypti* LIRAa) [13,17].
		16. Health System (0.974)	Variables of investment (national and local) and performance of Brazil Health System in municipalities, prenatal care and normal delivery, primary and higher levels of care coverage and public health actions. We included as one of the main public health action a very detailed documentation of vaccination about the number of doses (as a proxy of the strength of this intervention) and coverage (width of intervention over beneficiary population) for 2015 and 2016 [15,16].
Health outcomes		17. Vector Borne Diseases (0.677)	Severe dengue, Malaria and Chagas disease cases [15,16].
		18. Other Health Results (0.965)	Infant and childhood mortality, and incidence rate of congenital syphilis. Different poisoning incidence according to municipality of exposure, notification and residence [18]. Place of exposure hints direct contact with toxic substances, while place of residence and notification correspond to the administrative process.
		19. Microcephaly Surveillance	Surveillance of microcephaly incidence, including incoming and investigated microcephaly cases, and confirmed/discarded ZVI cases during the first semester of 2016 [19].
		20. Stillbirths Surveillance	Surveillance of stillbirth incidence, including incoming and investigated stillbirth cases, and confirmed/discarded ZVI cases [19].

## Data Availability

Aggregate data from this study is readily available by contacting the corresponding author. All data is stored within a customized, secure cloud environment in a data lake maintained by The Ohio State University.

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
