# Peer review of "Big Data to Knowledge Analytics Reveals the Zika Virus Epidemic as Only One of Multiple Factors Contributing to a Year-Over-Year 28-Fold Increase in Microcephaly Incidence"

_ijerph, 2022, doi:10.3390/ijerph19159051_

Round 1
Reviewer 1 Report
Main findings of the study
The study investigates the understanding about how simultaneous and interdependent factors interacted to determine the incidence of Zika-related microcephaly in Brazil.
Limitations and strengths of the study.
The strengths of the study are: a) the use of innovative technologies (Big Data) to address that the socioeconomic-political context scale had almost identical patterns of association with the m-ZIKV+- and m-ZIKV— subnetworks, b) to show the patterns of vulnerability, low education, income, and racial inequalities seem to be the strongest candidate factors for further action on microcephaly, in Brazil.
The main limitation of the study is the diagnostic failure to discern Zika infection during the epidemic and a possible underestimation of the size of Brazil’s ZIKV+ microcephaly incidence, according to the results. Despite controlling for this variable, it is likely that further studies with data on the prevalence of the Zika infection will be necessary to test the ecological model proposed by the authors. An important limitation of the study was the use of secondary data and the authors should highlight this limitation.
The abstract objectives are slightly different from article objectives. It is recommended that they are aligned.
Comment on the methods and results
The methods are innovative and complex, mainly the using of data mining. However, the method of the manuscript was not enough to understand the statistical parameters used. The Supplementary Materials of Materials and Methods needs to be moved to the manuscript, mainly the description about data collection, integration, and preprocessing.
These information are very important to understand the models e statistical analysis that were "trained" to identify how simultaneous and interdependent factors interacted to determine the incidence of Zika-related microcephaly in Brazil. In this sense I also recommend include the Fig.S1. Methods Flowchart into the manuscript.
All figures are illegible, mainly the figure 8 (the summary of the analysis). It is necessary to improve all of them.
Into the discussion section could add the consequences of the use of this innovative technology in leading to better support to reveal that the Zika virus epidemic is only one of multiple factors contributing with the increase of microcephaly incidence in Brazil.

Author Response
REVIEWER 1
Concern: The main limitation of the study is the diagnostic failure to discern Zika infection during the epidemic and a possible underestimation of the size of Brazil’s ZIKV+ microcephaly incidence, according to the results. Despite controlling for this variable, it is likely that further studies with data on the prevalence of the Zika infection will be necessary to test the ecological model proposed by the authors. An important limitation of the study was the use of secondary data and the authors should highlight this limitation.
Response: We agree with the reviewer that further study is needed to test the ecological model and that the use of secondary data has limitations. However, the unique capability of using the Public Health Exposome framework allowing the interrogation of disparate databases to examine factors contributing the cumulative risk and multicausality of microcephaly beyond Zika infection is innovative. Furthermore, while we agree that further studies are needed, primary data collection typically obtained through comprehensive environmental epidemiologic cohort or case control studies, are often difficult to carry out because of the limited public health and research infrastructure in the remote areas with the highest number of Zika cases. Also, these studies would be very expensive, and the findings would be delayed which would postpone time-sensitive mitigation strategies, interventions and/or control measures. Such interventions are particularly time sensitive to mitigate neurotoxicant exposures. We have modified the text in the last paragraph to clarify and enrich the discussion regarding limitations.
Concern: The abstract objectives are slightly different from article objectives. It is recommended that they are aligned.
Response: We thank the reviewer for pointing this out and have edited the abstract to align with what was done in the study.
Concern: The methodology in the submitted manuscript was not enough to understand the statistical parameters used. The Supplementary Materials and Methods needs to be moved to the manuscript, mainly the description about data collection, integration, and preprocessing.
Response: We appreciate the reviewer’s appreciation for the need to have this methodology narrative in the main body of the manuscript and have placed the data collection, integration, and preprocessing sections in the main body of the manuscript. Additionally, we have included Figure S1 the methods flowchart in the methods sections of the manuscript. Please see below the revised narrative and figure that was inserted into the METHODS section.
METHODS
A complex network of interdependent determinants of ZVI-microcephaly was generated by inferring associations between variables describing diverse factors and the incidence of microcephaly attributed to ZVI in Brazil municipalities during the first semester of 2016. This network provided the context to extract sub-networks of factors associated to ZVI+-microcephaly and controls that allowed to perform comparative inferential analysis beyond a descriptive approach. Figure S1 summarizes the process and analysis.
Data collection, integration, and preprocessing
Secondary data was collected secondary open data from Brazil`s official websites about factors according to the WHO conceptual framework of the social determinants of health (SDH)28 summarized in Table 1. Additionally, data in Table 1 was included for ZVI-related microcephaly incidence from the open data initiative of the Brazilian system of public health surveillance and other health results.
By the standard identifier of the various Brazilian municipalities we integrated an initial set of more than 700 variables for 5570 municipalities29. Pre-processing these continuous, discrete, and binary variables included identifying and replacing missing values according to particular customary processes of collection and selecting n=5,565 complete cases (population of municipalities). All binary variables of the context determinant were included. Pairwise collinearity and concurvity detection assisted variable selection of the remaining numeric variables (651) of structural and intermediary determinants, and health results. Collinearity criteria included Pearson correlations, their Fisher-z transform and VIF (variance inflation factors) beyond 2, 2.5 and 3 standard deviations, and 2 and 3 IQR (Interquartile range). Concurvity criteria were Spearman correlations beyond 2 and 2.5 standard deviations and IRQ. We also determined the distributions, thresholds, and matrices to visualize different patterns of criteria that eased team discussions to downsize the number of variables to 382.
By Cronbach’s alpha, we validated construct reliability of variables grouped in the four scales (context, structural, intermediary determinants, and health results) that arrayed 18 determinants. For each determinant, Table 1 describes the variables included, their source, Cronbach’s alphas and the colors assigned for further visualization of results.
Network generation
We obtained the general structure of the complex network of associations among the m=382 final variables by inferring throughout the entire data all possible linear Pearson and monotonic Spearman higher-order partial correlations for interval/ratio variables and equivalent point-biserial correlations for binary variables (R package pcor.30 As no variable was previously labeled as dependent or independent, the undirected character of partial correlations fulfils the exploratory and objective scope of this study.
As higher-order partial correlations (pcor) control associations between each pair of variables by removing the effects of the remaining full set of m-2 variables throughout the entire data, we accounted for, and reduced confounding and spuriousness induced by any third variable. To quantitatively substantiate the global effect of confounding and the consequent reduction of spurious associations that resulted from using partial correlations, we also tested for differences between Fisher transformations of higher-order partial and zero-order coefficients.
Significant differences between distributions of zero-order and partial correlation coefficients (t-test p<0.001, 95% CI [0.1287-0.1224]) made explicit the presence of confounding throughout the data. A mean difference with wide standard deviation of 0.12±0.61 between both types of correlation endorsed our choice of higher-order partial correlations to infer the network structure.
Hartemink’s mutual information method [R package bnlearn]31 was used to reconcile different number of levels on the distributions of (mx(m-1))/2 = 72,771 Pearson and Spearman partial correlations and their corresponding p-values (non-corrected and corrected by family wise error rate methods of Bonferroni and Holm-Šídák, and by Benjamini–Hochberg false discovery rate procedure) to find objective and meaningful thresholds. As all the strongest Spearman coefficients were included in lower and higher tail categories of the Pearson coefficients distribution, we maintained this last category to define final seven thresholds and retain meaningful intervals corresponding to the upper and lower two as the strongest positive [0.812,1.000], strong positive [0.509,0.812), strong negative (-0.711,-0.442], and strongest negative [-1.000,-0.711].
The resulting network obtained was composed of 99.73% of the initial variables as nodes and 36,400 statistically significant and meaningful associations as links. Customary indices in Table S2 (supplementary data) summarize the overall properties of this general network structure. According to KS test, with a large left tail, a negatively skewed degree distribution did not fit Normal, Poisson or other functions, but an R2=0.394 showed a significant but weak fitting to a power law function. In this way, the dense general network was still asymmetric allowing for some sort of differential organization.
As the network is formed by controlled associations corresponding to statistically based confounding-free relationships, different types of triads and related indices such as Watts Strogatz and clustering coefficients (CC1 and CC2), provide clues about interaction, which is relevant for epidemiological analysis. In this way, variables part of one edge triads shows pairwise associations independent of others, intransitive triads correspond to multiple independent associations and transitive triads display synergistic or antagonistic interactions. In the network, transitive triads prevail showing that the multiple factors associated to any outcome node simultaneously act and even modify effects of each other.
Topological analysis required further reduction of the network density looking for an optimal consistent structure. By maximizing the difference between clustering coefficients of the general network and a generated random ErdÅ‘s–Rényi model, we identified a threshold of |0.76| absolute partial correlation. This thresholded network captures associations with strongest transitive and therefore interdependent interactions. In addition, by ‘Graphlet Correlation distance, we used the 2- to 5-nodes graphlets32, which correspond to small subgraphs, to best capture the local wiring patterns (structures) around the nodes in networks, and we measured the structural distance between two networks using their graphlet correlation distance33 in which, to reduce the effect of structural decencies between graphlets, we only considered the 58 non-redundant 2- to 5-node graphlet orbits (GCD-58). We also computed the topological distance between the general network at various thresholds and a set of network models (Figure S1) found an additional threshold at |0.65| absolute partial correlation for the maximal distance to ErdÅ‘s–Rényi model. The resulting thresholded network captures more complex interactions among groups of 3-5 factors and outcomes. Indices were computed also for thresholded networks (Table S2).
For network visualization, by the energy-based Kamada-Kawai algorithm that includes link weights, we obtained two-dimension coordinates in two levels among determinant factors and among nodes within factors. At the determinant level, the weights used were avg-pcor2, and at node level weights were within-determinant pcor. Node colors were assigned previously according to Table S1.
We assessed the global significance and validity of the general network by statistical tests between these customary indices and equivalent parameters of sampling distributions obtained by Monte Carlo simulation (MCS) of 1,000 random graphs according to the Bernoulli variant of the ErdÅ‘s–Rényi model based on the observed average degree.34 Table S2 shows the confidence intervals of different indices as an estimate of the gap between the general network and random graphs. All network indices were significantly different to the corresponding parameters of the MCS-based sampling distribution, rejecting that the network structure obtained by higher-order partial correlations was due to chance. Additionally, average of the Grapplet Degree Distribution (GDD)35 of the general network to the ErdÅ‘s–Rényi model was 0.66±1.84e-5.
Construct reliability of determinants of health (node groups in Table S 1), which was verified during data preprocessing by Cronbach’s alpha, was also confirmed by canonical-correlation analysis that identified statistically significant relationship between determinants and the graphlet orbits representing the local topology of the network at both |0.76| and |0.65| thresholds.33 Additionally, canonical correlation analysis in red, green plots, displays the complexity of interactions among determinant factors (Figure S3). According to the 1-cannonical variable from the network thresholded at 0.65, all pooled microcephaly surveillance incidence distributions are anticorrelated with ‘education’, and partially with ‘social networks’ and ‘biological factors’, and ‘occupation’ acts as a broker or mediator. Showing closer interaction as part of square structures, intermediate factors such as health system, material circumstances, other health results and childhood development are predominantly closer to microcephaly surveillance. Income and social class are also close, but the first is anticorrelated. The 2-cannonical variate shows that microcephaly surveillance, income, social networks, and macroeconomic policy belongs to interdependent square structures, while race/ethnicity, occupation, biological factors can act as mediators, and material circumstances, vector borne diseases and health results are more peripheral. As none of the canonical analysis with the network thresholded at 0.76 included microcephaly surveillance, results showed how the remaining factors are interdependent and shape profiles of well-being in municipalities.
Inferential network analysis by subnetwork comparison
Testing the system of composite hypothesis relied on extracting four subnetworks from the full and thresholded (0.65 and 0.75) networks presented before. Subnetwork extraction relied on a k-neighborhood procedure, which holds the neighbor nodes closely associated with specific target nodes/variables. The target node of municipal distribution of 1,810 cases of microcephaly with confirmed ZVI defined the m-ZVI+-subnetwork; the node of municipal distribution of 3,924 microcephaly cases negative for ZVI defined the control m-ZVI--subnetwork. Target variables of municipal distribution of 117,326 low birth weight (LBW) cases and 863,153 births (B) defined the additional control subnetworks. Table S 3 presents the indices describing each subnetwork.
By composite hypothesis testing using various statistical parametric and non-parametric methods to account for similarities and differences, we compared components and topology of subnetworks and their association patterns. Compared components included vectors and distributions of neighbor nodes, unweighted and weighted links, and customary indices. Statistical tests included coefficients and p-values of zero-order Pearson, Spearman, Kendall and Phi correlations, and Student’s t, Wilcoxon signed rank, McNemar and Kolmogorov-Smirnov (KS) for corresponding independent and dependent samples.36 We also identified the type of function that better described each subnetwork degree distribution by computing R2 determination coefficients with different reference models. For comparing subnetwork topologies, we used Graphlet Correlation Distance (GCD) with GCD-58 already described.
To interpret the detailed meaning of results of composite hypothesis testing between the four subnetworks across scales, determinants and single variables, we focused on higher-order squared partial correlations (pcor2) and their averages, which are known measures of effect size that account for the proportion of variance involved in each association without effects of other variables (10). For scales and determinants, we computed averages (avg-pcor2) via Fisher’s z-transformation of pcor. However, pcor and zero-order correlations (zcor) nuanced our findings that were mainly based on pcor2. In this way, pcor similar to zcor hinted spurious associations, and pcor higher than zcor indicated the presence of suppressor variables.36-37
Additionally, by color bands that were extracted from the corresponding heat maps matrices of zcor, pcor and pcor2 for full and thresholded subnetworks, we complemented the analysis with visual profiles or fingerprints of direct connections for each subnetwork (Supplementary material, S2). In pcor2 bands, darker blue shade designates high link values, while lighter shades indicate low link values. Fingerprint bands for zcor are represented by a continuous ascendant spectrum between red, yellow, and green, while the colors of pcor fingerprint bands follow the same four intervals for positive and negative strong and strongest links, as shown in network visualizations.
In Supplementary Materials, we present additional data used to collect data in accordance with the open policy initiative related to Zika Virus Disease and analysis of the network by testing composite hypotheses using BD2K advanced network science. See Table 1 that details sources of available data that were utilized.
Concern: All figures are illegible, mainly the figure 8 (the summary of the analysis). It is necessary to improve all of them.
Response: We appreciate the reviewer’s comments on this. We did not intend for the figures to be illegible and have requested the associate editor to use a full page for the 3-main figures in the manuscript which will be Figures 2, 3 and 4.
Concern: Into the discussion section, the authors could add the consequences of the use of this innovative methodology in leading to better support to reveal that the Zika virus epidemic is only one of multiple factors contributing with the increase of microcephaly incidence in Brazil.
Response: We thank the reviewer for this comment and will point the reviewer to our discussion on this point.
The recently described Public Health Exposome (PHE) framework with Big Data to Knowledge (BD2K) analytics is a novel environmental public health framework that could assist local urban public health and rural municipalities in a post evaluation of Zika infections in Brazil. The framework could be used to predict and model relationships between environmental and socio-demographic variables and adverse health outcomes.28 In this instance and going forward, the PHE would be available to assist public health officials in northeast Brazil with integrating, partitioning, and clarifying pockets of increasing prevalence in low birth weight and pre-term births from LBW-subnetworks containing both m-ZIKV+ and m-ZIKV-. Additionally, the PHE framework would also inform public health officials as to how chemical stressors such as components of ambient pollution negatively impact and potentiate adverse pregnancy outcomes. We now know that there is sufficient evidence in 2020 to support the fact that exposure to chemical stressors causes important adverse health outcomes that result in premature death, including ischemic cardiovascular disease, stroke, respiratory infections, chronic obstructive pulmonary disease, and lung cancer. By using our BD2K framework and analytics to integrate environmental science with population-based study designs, it has the potential to contribute novel and innovative theories, mitigation strategies and interventions to dampen negative health outcomes by assessing risk trajectories from exposures to chemical and non-chemical stressors.
Figure S1.
Figure S1. Methodology Flowchart. Subsequent to data collection, integration, preprocessing and network generation, the third step of model analysis consisted of extraction and comparison of 4-subnetworks to test three morbidity hypotheses. Subnetworks are identified by letters and comparisons between networks by lower-case roman numerals.
Reviewer 2 Report
Cifuentes et al. present interesting evidence for non-Zika related microcephaly and low birth weight conditions in Brazil during the time of the Zika outbreak there. Using a big data approach, they analyze a multitude of variables and ultimately conclude that microcephaly is multifactorial, such as exposure to heavy metals, rather than related primarily to ZIKV infection during this time. This is a well constructed analysis and is a worthwhile addition to the ZIKV literature.
Figure 1 appears to be duplicated side-by-side and is difficult to read. This makes it difficult to evaluate.
Materials and methods mentioned semesters. Is this the intended wording? If so, please define.
Lines 145, 152, 154, 155, 170, 172, 195, 198, and 204 have an error for the reference when referring to figures/tables. Please correct.
References in the main body seem to not match with the bibliography. For example, in line 184, Schwartz is listed as citation 28, but in the bibliography, is number 30. References need to be corrected.
Author Response
REVIEWER 2
Concern: Figure 1 appears to be duplicated side-by-side and is difficult to read. This makes it difficult to evaluate.
Response: We apologize for this apparent oversight and have corrected this. Please see response to Reviewer 3.
Concern: Materials and methods mentioned semesters. Is this the intended wording? If so, please define.
Response: We have removed the mentioning of semesters and apologize for this.
Concern: Lines 145, 152, 154, 155, 170, 172, 195, 198, and 204 have an error for the reference when referring to figures/tables.
Response: We thank the reviewer for pointing this out and have corrected all errors in each of the lines and did so by going back to the original manuscript to ensure that these errors do not show up in the Word file of the revised manuscript.
Concern: References in the main body seem to not match with the bibliography. For example, in line 184, Schwartz is listed as citation 28, but in the bibliography, is number 30.
Response: We thank the reviewer for pointing this out and again, have rectified this issue by by going back to the original manuscript making the corrections to ensure that these errors do not show up in the Word file of the revised manuscript.
Reviewer 3 Report
This is a potentially interesting manuscript, dealing with additional variables that may have impacted the prevalence of microcephaly after zika infection during pregnancy. However it is impossible now to evaluate properly this manuscript, since the figures are in very low resolution and a definitely cannot read. Take Figures 4-7, for example. This is a key figure to understand the results, but it is really impossible to read it.
Other important aspects that should be improved:
1. The authors say that the geographical distribution of ZIKV infection and microcephaly didn't match. However the Figure 1 , where the authors present as the basis for their assumption did NOT show this - this figure ONLY presents distribution of ZIKV induced microcephaly and NOT the distribution of the infection itself. Moreover, the distribution presented by the authors apparently agree with the distribution of ZIKV infection to my best knowledge of the situation in Brazil. I agree that some modifiers should be present because in areas with similar occurrence of ZIKV there were different incidences of microcephaly. Therefore the authors should show, in this Figure, one map the ZIKV distribution and in the other map the ZIKV induced microcephaly.
I emphasize, however, that this is a very interesting and meaningful approach, but needs to be improved to present clearly the results.
Author Response
See attached Word file.
